# Low Expression of miR-375 and miR-190b Differentiates Grade 3 Patients with Endometrial Cancer

**DOI:** 10.3390/biom11020274

**Published:** 2021-02-13

**Authors:** Miłosz Pietrus, Michał Seweryn, Przemysław Kapusta, Paweł Wołkow, Kazimierz Pityński, Gracjan Wątor

**Affiliations:** 1Department of Gynecology and Oncology, Jagiellonian University Medical College, 31-501 Krakow, Poland; milosz.pietrus@uj.edu.pl; 2Center for Medical Genomics OMICRON, Jagiellonian University Medical College, 31-034 Krakow, Poland; panpit@gmail.com (M.S.); przemyslaw.kapusta@uj.edu.pl (P.K.); pawel.wolkow@uj.edu.pl (P.W.); 3Department of Pharmacology, Jagiellonian University Medical College, 31-531 Krakow, Poland

**Keywords:** endometrial cancer, miRNA, EC grade, NGS, biomarkers

## Abstract

Endometrial cancer (EC) is treated according to the stage and prognostic risk factors. Most EC patients are in the early stages and they are treated surgically. However some of them, including those with high grade (grade 3) are in the intermediate and high intermediate prognostic risk groups and may require adjuvant therapy. The goal of the study was to find differences between grades based on an miRNA gene expression profile. Tumor samples from 24 patients with grade 1 (*n* = 10), 2 (*n* = 7), and 3 (*n* = 7) EC were subjected to miRNA profiling using next generation sequencing. The results obtained were validated using the miRNA profile of 407 EC tumors from the external Cancer Genome Atlas (TCGA) cohort. We obtained sets of differentially expressed (DE) miRNAs with the largest amount between G2 to G1 (50 transcripts) and G3 to G1 (40 transcripts) patients. Validation of our results with external data (TCGA) gave us a reasonable gene overlap of which we selected two miRNAs (miR-375 and miR190b) that distinguish the high grade best from the low grade EC. Unsupervised clustering showed a high degree of heterogeneity within grade 2 samples. MiR-375 as well as 190b might be useful to create grading verification test for high grade EC. One of the possible mechanisms that is responsible for the high grade is modulation by virus of host morphology or physiology.

## 1. Introduction

Endometrial cancer (EC), the most common gynecological tumor in developed countries is characterized by a growing incidence rate and is frequently symptomatic at an early stage [1]. To plan an optimal therapy in patients with EC, clinicians need to know the most important phenotypic feature after they establish the tumor stage, i.e., tumor grade. The identification of cases with grade 3 tumors may negatively affect patient prognosis. The 5-year distant recurrence rates were 3%, 8%, and 25% for grades 1, 2, and 3, respectively (*p* < 0.001); and the 5-year disease-specific survival rates were 97%, 94%, and 76%, respectively (*p* < 0.001) [2]. According to the current risk assessment/treatment guidelines, the differentiation between grades 1 and 2 is less important, because it does not determine inclusion in specific prognostic risk groups and potential use of adjuvant therapy, as the requirement for this should not be decided merely based on the patient’s grade [3]. Currently, only The International Federation of Gynecology and Obstetrics (FIGO) and Union Internationale Contre le Cancer (UICC) system with three possible grades is used for the histopathological grading of gynecological cancers [4]. Assignment to the appropriate grade is based on the percentage of solid non-squamous, non-morular tumor cells during histopathological evaluation, with grade 3 assigned to cases with over 50% of the tumor cells. However, this assignment is affected by the inter-observer variability, especially in high-grade EC, when the diagnosis is based on histomorphology alone [5,6]. The availability of molecular methods encourages to support EC diagnosis with genomic and transcriptomic data.

Most EC cases are low grade (grades 1 or 2) and are cured by hysterectomy with sentinel lymph node biopsy in early stages with favorable prognosis, whereas high-grade (grade 3) EC cases have a significantly worse prognosis. Moreover, current pathological classification is limited in reproducibility and prognosis [7].

MiRNAs are small RNAs, 22 nucleotides long, that base-pair to target mRNA through an RNA-induced silencing complex (RISC). Each miRNA is predicted to regulate up to hundreds of targets that are involved in a broad range of processes, including proliferation, differentiation, and apoptosis [8]. Currently, growing evidence suggests that miRNA gene expression is dysregulated in cancer and its signature can be used to classify the tumor. Moreover, miRNAs engage in traits that are hallmarks of cancer and can act either as oncogenes or as tumor suppressor genes [9]. Another important feature of miRNAs is that they are relatively stable, which makes them ideal candidates for biomarkers [10].

Accumulating evidence suggests that a specific miRNA signature differentiates particular subtypes of tumors. For example, two histological subtypes of gastric cancer (intestinal and diffuse) were accurately distinguished using miRNA signature [11]. Another example is the accurate distinction of renal cell carcinoma (RCC) from benign-behaving renal oncocytoma using miRNA expression profiling [12]. The global study of the Cancer Genome Atlas cohort (TCGA) data (19 tumor groups) using spectral clustering of similarity network fusion analyses of molecular platforms (including miRNA sequencing data) showed high concordance (89%) with tumor histology [13].

So far, most of the studies conducted on EC have focused on miRNA expression profiles. Most of them are specifically aimed at prognostic factors such as lymph node involvement, lymphovascular space invasion (LVSI), overall survival (OS), and recurrence-free survival (RFS). Unfortunately, most of the gene expression analyses of miRNA are based on quantitative polymerase chain reaction (qPCR) or on microarray data (10 microarrays vs. two next generation sequencing (NGS)), and have rarely focused on NGS [14]. Using qPCR or microarray we are limited to examining only known sets of miRNAs, usually bigger in microarray experiments whereas NGS gives us an opportunity to study the whole miRNome. NGS was applied by Tsukamato and colleagues to find a specific endometrioid endometrial carcinoma (EEC) associated miRNAs in tissue and plasma [15]. The goal of their study was to use a set of miRNAs for early detection as well as for follow-up tests. Another group, led by Xiong, used NGS to investigate the potential molecular mechanism underlying the pathogenesis of EEC. They simultaneously sequenced total mRNA and small RNA of only three pairs of stage I EEC and adjacent non-tumorous tissues [16].

Histologic classification remains a gold standard for diagnostic and prognostic classification of cancers. However, there is significant inter-observer variability in the pathologic diagnosis of high grade carcinomas [17]. For example, Gilks and colleagues found only 62.5% (35/56) cases with diagnostic consensus among three reviewers about the proper subtype of high grade EC [18].

Therefore, in this study, we aimed to find the differences between well-differentiated (G1), medium-differentiated (G2), and poorly-differentiated (G3) EC based on the expression profile of miRNAs. In particular, we wanted to test whether the miRNA profile is able to reclassify individual samples, as well as assess a degree of homogeneity in particular groups.

## 2. Materials and Methods

### 2.1. Patients

In this study, we used material obtained from patients who were hospitalized in the Gynecology and Oncology Department of the Jagiellonian University Hospital in Cracow during 2016–2017 and who qualified for the surgical treatment for EC. Patients whose tumor cells were obtained from the uterus during their previous biopsy of the uterine cavity and those in the age group of 35–80 (eligibility criteria) years were included in this study. Patients with other cancers in the past or present, as well as those who had undergone prior radio and chemotherapy and also those with family links with any other patients were excluded from this study.

### 2.2. Study Approval

The study protocol was approved by the Jagiellonian University Bioethics Committee (agreement no. 122.6120.283.2015; 17 December 2015) and informed consent was obtained from each patient before the surgery.

### 2.3. RNA

RNA was extracted from 50–100 mg of the tissue sample (stored in RNA-later solution) using mirVana miRNA Isolation Kit (Thermo, Ambion, Carlsbad, CA, USA) according to the total RNA isolation procedure. Prior isolation tissue was disrupted in TissueLyser LT (Qiagen, Hilden, Germany), with zirconium beads, for two sets per 3 min with 1 min rest interval and at 50 rpm. The quality of RNA was determined using TapeStation (Agilent, Craven Arms, UK), whereas the concentration was measured using Nanodrop (Thermo). The median RNA integrity number (RIN) was 7.3 (range 3.9–8.8).

### 2.4. NGS

A small RNA library was generated using New England Biolabs chemistry. We used 100 ng of total RNA for adapter ligation. The library was enriched by 12 cycles PCR and size selected using BluePippin to cut out 141 bp fragments. We pooled six samples per single MiSeq run and set up 12 pM library pool on v2 (50 cycles, single read) cartridge. To increase the diversity of the library pool we added an artificial library (PhiX, Brea, CA, USA) at a final concentration of 20%.

### 2.5. Bioinformatic Pipeline and Data Analysis

Data were collected and stored on our local server. After demultiplexing, the quality of the generated data was checked using FastQC software [19]. After removing the adapter with Cutadapt [20], the trimmed reads were aligned to mirBase 22.1 [21] and counted using miRDeep2 software [22]. The raw read counts were imported to the statistical software R and processed using the “edgeR” pipeline [23]. In short, the results were normalized for library size, miRNAs with low counts (median below five reads) were removed from further analysis, and dispersions (common, trended, and tagwise) were estimated in the context of the generalized linear model. In all linear models, an independent variable that corresponds to tumor grade was implemented as a factor, therefore the differentially expressed genes were detected using contrasts between values of this variable. No additional covariates were taken into account to minimize the risk of overfitting, due to small sample size. A miRNA was considered differentially expressed at FDR < 0.05. The TCGA data were downloaded and processed using the ‘TCGAbiolinks’ package in R. As for the in-house cohort, the independent variable associated with tumor grade was treated as a factor. In all the differential expression analysis, the Benjamini–Hochberg multiple testing correction was applied.

### 2.6. QPCR

The most reliable miRNAs were validated using quantification PCR with intercalating dye—SYBR Green. Briefly, 20 ng of RNA was used to perform reverse transcription (miRCURY LNA^TM^ Universal RT microRNA PCR from Exicon). Generated cDNA was diluted 60 times and applied in triplicates to qPCR reaction (miRCURY LNA^TM^ miRNA SYBRGreen PCR) with specific primer set and reference gene SNORD48 (Qiagen/Exicon, Vedbaek, Denmark). The reaction mixture was carried out on CFX384 (BioRad, Singapore) thermal cycler with protocol according to primer set provider (Exicon). Raw Cq values were processed using GenEx (MultiD, Gothenburg, Sweden) software. Statistical analysis was conducted in R (one way ANOVA with Tukey’s contrast) and GenEx (data conversion and box plot).

## 3. Results

### 3.1. Study Group

The study group included 24 patients with EC. Based on the histopathological evaluation, our cohort included 10 patients with grade 1, 7 patients with grade 2, and 7 patients with grade 3. Except for one patient, in whom lymph nodes were absent in the post-operative material, the rest of the patients did not have lymph node metastases. The early stages (IA and IB) prevailed (Appendix A, Table 1).

### 3.2. MiRNA Profiling

The miRNA profiling of 24 EC tumor samples was performed in four MiSeq runs. Overall we obtained more than 27 M reads which gave us on an average 1,135,056 reads per sample. We identified 1128 miRNA transcripts with at least five reads gene expression. The most abundant transcript was miR-21-5p with overall read count of 4,006,770 (14.7% reads). It is noteworthy that the five most abundant transcripts (miR-21-5p, miR-148a-3p, miR-143-3p, miR-10a-5p, and miR-10b-5p) consumed almost 46.8% of all reads.

### 3.3. Data Cleaning

For exploratory analysis we used the multidimensional scaling (MDS) plot method. The analysis did not reveal any obvious clustering within the studied group. However, we can see an outlier, sample no. 24 (Appendix A). This sample had mixed endometrioid carcinoma with undifferentiated carcinoma type, as well as the most advanced stage (III) in this group, and was removed from further analysis.

### 3.4. Differential Expression Profile

In this study, we detected differentially expressed miRNAs between G2 and G1, G3 and G1, and G3 and G2 (Appendix A). We found the largest number of differentially expressed miRNAs within the G2 to G1 comparison (50 transcripts), as well as within the G3 to G1 comparison (40 transcripts). The least number of differentially expressed miRNAs was found for the G3 to G2 comparison (among the three transcripts, miR-375, miR-214, and miR190b, two of them were confirmed using external TCGA data—see next paragraph). The most statistically significant (in the G3 to G2 comparison) was miR-375, whose reduced expression has been reported in ECs with positive lymph node status [24]. Furthermore, miR-214-3p has also been described as an important regulator of Wnt/ß-catenin pathway in EC cells [25]. The third transcript, miR-190b, downregulated in our data, was included by Wu and colleagues to the panel of biomarkers to predict survival in EC [26].

Our data show that most miRNAs are downregulated with increasing grades of EC (volcano plots, Figure 1). When comparing the expression leve ls of miRNAs in patients with G2 versus G1 EC, we found that 40 transcripts were downregulated and 10 were upregulated (range of logFC: −3.45 to 2.51).

### 3.5. Possible Grade Signature

To further evaluate the discriminative power of the differentially expressed miRNAs we applied a random forest model to both estimate the minimal number of discriminative features (via cross-validation) and to find the set of most informative miRNAs. To accomplish this we used our in-house data, and therefore, we treat this piece of analysis as rather speculative, thus requiring further validation. We initially considered 72 differentially expressed miRNAs and by means of cross-validation, we estimated that the number of features that gives the lowest error rate is about 10. Subsequently, in a random forest model with all 72 features, we evaluated the mean decrease in Gini index [27] to select the set of the 10 most discriminative miRNAs. Finally, we trained a random forest with only 10 predictors (hsa-miR-652-3p, hsa-miR-4510, hsa-miR-190b, hsa-miR-421, hsa-miR-941, hsa-miR-1307-3p, hsa-miR-375, hsa-miR-345-5p, hsa-miR-99b-3p, and hsa-let-7b-3p) and obtained a confusion matrix with likelihood error rate (Table 2).

### 3.6. Validation of Results Using External TCGA Data

We can clearly see that in the case of individual miRNAs, our results are consistent with those available in the literature. However, our cohort study (JUMC, Jagiellonian University Medical College cohort) has a limited number of samples; therefore, we validated our results using other independent EC data set from the TCGA initiative. We chose only low and high grade EC. Overall, we tested 407 TCGA samples—G1 (*n* = 97), G2 (*n* = 119), and G3 (*n* = 191). NGS profiling of these samples was conducted using Illumina technology and the Illumina TruSeq Small RNA library preparation system. We grouped TCGA samples in the same way as we group them for cluster analysis (see next paragraph): G1 + G2 samples against G3 samples (low grade vs. high grade) and G1 against G2 + G3 samples. As a result, we obtained 176 differentially expressed miRNAs for the first comparison and 131 differentially expressed miRNAs for the second comparison. Then, we extracted overlapping genes—G1 or G2 versus G3 (6 out of 10 genes, 60%) and G1 versus G2 or G3 (20 out of 65 genes, 31%) (Figure 2). Despite the small number of samples in our cohort, we obtained 60% and 31% gene overlap with external data. In addition, we confirmed the significance of miR-375 and miR-190b in recognizing grade 3 EC. Moreover we performed a technical validation of these miRNA using qPCR. In grade 3 patients, on average, the logarithmic fold change (logFC) of miR-190b and miR-375 were −3.0437 (*p* less than 0.001) and −4.392 (*p* = 0.00289), respectively, compared to grade 2 patients. This measurement accurately confirmed the results from RNAseq (−3.3348 for miR-190b and −4.3429 for miR-375).

The main goal of our research was to find an miRNA pattern for a specific grade of EC. We attempted to find an independent predictor of grade, based only on miRNA profile. This does not imply that we dismiss the utility of previously identified predictive factors. We have found that the expression of specific miRNAs may aid the classification of tumor according to grade. However, we performed multivariate analysis with other predictors added one by one to the model which included miRNAs such as height, weight, menopause status, tumor invasion, age, and ethnicity. We found out that miR-190b is better predictor than miR-375 and remained statistically significant in relation to all added predictors (miR-190b height, *adj.p [adjusted p value]* = *0.000067*; weight, *adj.p* = *0.000028*; menopause status, *adj.p* = *0.0043*; tumor invasion, *adj.p* = *0.00012*; age, *adj.p* = *0.000017*; ethnicity, *adj.p* = *0.000028*) whereas miR-375 remained significant only with tumor invasion (*adj.p* = *0.0085*).

### 3.7. Clustering of Samples in the Grading Context

It is noteworthy that indeed the changes in expression levels between G3 and G2 are the most subtle (Figure 3 and Figure 4). We performed unsupervised clustering of 23 samples to check the fit of individual cases to their respective grades, known from the gold standard histopathologic evaluation. Unfortunately, the sample size in our cohort is not sufficient to draw reproducible conclusions by means of unsupervised learning methods. Therefore, we combined grades 1 and 2 against grade 3 and grades 2 and 3 against grade 1 for differential expression experiments. We used the set of differentially expressed miRNAs from the analysis of linear models for unsupervised clustering (Figure 3 and Figure 4 show heatmaps from the top 30 differentially expressed genes). The discrimination analysis of G1 versus G2 or G3 gave as a clear coherent cluster of six samples with G1 and a mix of the rest of the 17 samples, without any distinct pattern for G2 and G3 (Figure 3). The remaining G1 samples (4 from 10) were mixed with G2 samples (2 G1 samples), G3 samples (1 G1 sample), and G2-G3 samples (1 G1 sample). The second discrimination analysis (of G1 or G2 vs G3) gave as two clear coherent clusters: a cluster of 5 G1 samples and a cluster of 5 G3 samples (Figure 4). The remaining samples were dispersed without any specific pattern. In summary, our miRNA expression analysis confirmed the large differences between grades 1 and 3 EC.

Based on cluster analysis, we can indirectly estimate the level of grade homogeneity. In the case of grade 1, the heat map shows us a cluster of six samples for overall 10 samples, which means 60% (6/10) of homogeneity was obtained (Figure 3), whereas grade 3 samples have 84% (5/6) homogeneity (Figure 4).

Another concern is which of the miRNAs are the best for differentiation between specific grades. Based on our gene expression analysis of miR-1-3p, we could capture a homogenous group of grade 1 samples. However, other grades did not have specific single miRNA but rather have a group of miRNAs.

### 3.8. Gene Ontology of Predicted Targets

Subsequently, we asked which of the biological processes is responsible for the lack of differentiation of the EC cells. Based on the discovered miRNAs, we predicted their target miRNAs. We choose a set of unique miRNAs from differential expression results. Overall there were 72 unique miRNA from all comparisons (G3 vs. G2, G3 vs. G1, and G2 vs. G1). We carried out gene ontology analysis for all comparisons and for separate groups (Appendix A). The following three are the most significant processes in which miRNAs are indirectly involved: (1) positive regulation of transcription by RNA polymerase II (GO:0045944), (2) negative regulation of cell differentiation [GO:0045596], and (3) modulation by virus of host morphology or physiology [GO:0019048].

## 4. Discussion

According to our knowledge, this is the first time when tumor miRNome has been used to discriminate EC grades. We found down-regulation of miR-375 and miR-190b between grade 3 and grade 1–2 and up-regulation of miR-214 between grade 3 and grade 2. This achievement can be used to recognize or to help to classify samples to a specific grade. After the tumor stage, the tumor grade is the most informative predictor of disease severity [3]. Using NGS-based miRNAs profiling of tumor samples, we found differences between each grade of EC, which was the primary goal of this study. Moreover, we have confirmed heterogeneity of G2 samples and have discovered three miRNAs (miR-375, miR-214, and miR190b) that discriminate this grade from grade 3. In addition, we hypothesize, based on our results, that diagnostic tests combined with the 10 most discriminative miRNAs, might be useful in assessing, with better power, the specific EC grade than traditional pathology test.

Accumulated genomic and transcriptomic data has reached a sufficient level for use in improving diagnosis and stratification of patients [28]. A huge effort of the TCGA consortium resulted in the development of many new cancer typing systems, e.g., in colorectal cancer, breast cancer, glioblastoma, and EC [29,30,31,32]. The primary advantage of molecular typing is less dependence on the knowledge and experience of a pathologist. In addition, tests are less prone to human error and far more objective with a continuous measurement scale (e.g., gene expression analysis) rather that discrete results. The observed variable (molecule) can be different and its choice is dictated primarily by its ubiquity in cancerous tissue and the availability of the specific material.

Considering the above arguments, miRNAs seem to be the ideal molecules for testing cancer for diagnostic and prognostic purposes. These molecules are relatively stable, easy to manipulate and with sufficient tissue specificity [33]. They can be used as a diagnostic and prognostic biomarkers and as a potential therapeutic target [34]. These properties of miRNAs led us to better distinguish patients with a certain EC grade.

Historically, the distinction between low- (grades 1 and 2) and high-grade EC (grade 3) was the most difficult. This was also confirmed in our results. In this study, we found many more miRNAs differentiating G1 from G2 (50 transcripts) and G1 from G3 (40 transcripts) compared to miRNAs differentiating G2 from G3 (three transcripts). The three transcripts—miR-375, miR-214-3p, and miR-190b, have already been described in the context of EC metastasis or poor survival of patients. We successfully validated miR-375 and miR-190b using TCGA cohort.

Mir-375 was first identified from the murine pancreatic β-cell line and characterized as a pancreatic islet-specific miRNA but this transcript is widely present in various of tissues or organs and significantly down-regulated in many type of cancer cells—hepatocellular carcinoma, esophageal carcinoma, gastric cancer, head and neck cancer, melanoma, and glioma and rarely up-regulated, for example in breast cancer and prostate cancer. Generally, miR-375 acts as tumor suppressor, repressing many critical oncogenes such as PDK1, JAK2, IGF1R, and AEG-1 [34]. Less known, is miR-190b, although it is also engaged in promoting many cancers such as bladder carcinoma [35] or colorectal cancer [36] and found, as well, as a novel tumor suppressor in lung cancer during in vivo miRNA knockout screening [37].

A natural implication of this study is to answer the question of whether we can use this result to properly classify or reclassify EC grade. We are aware of the fact that this study group is too small to give a reliable biomarker and this was also not an intention of this study. However, we calculated, speculatively, using the random forest method, the sufficient number of miRNAs that could distinguish a particular EC grade. Our prediction suggests to use at least 10 miRNAs to correctly distinguish each grade (with a maximum likelihood of 14.3% of making an error in the case of grades 2 and 3). Therefore, this predictor would be far more specific in the case of high-grade EC compared to morphological examination, where 37.5% of the cases are probably incorrectly assessed [18].

Undoubtedly, help in diagnosis is just the beginning of using knowledge about molecular processes that drive tumor progression and fate. The ideal would be to provide a molecular stratification of patients before and during therapy. Currently, different type of molecular markers are used to predict poor progression-free survival (PFS). For example, aberrant (tested immunohistochemically) expression of p53 in cases with G1 or G2 is associated with poor survival [38]. On the other side, we could determine G 3 cases with better recurrence-free survival (RFS) if we applied molecular subtyping. From four molecular subgroups of grade 3 EEC, the POLE group (presence of POLE exonuclease [DNA polymerase epsilon catalytic subunit A] domain hotspot mutation) and MMRd (loss of mismatch repair protein expression) are an independent prognostic factor for better RFS [39].

The first attempts to use the miRNA profile to predict OFS, PFS, and RFS have also been made [40,41,42]. Tang and colleagues used TCGA data to establish a prognosis signature based on the hazard ratio (HR) of each miRNA. They enrolled seven miRNAs for OS prediction signature. Surprisingly, two of these, miR190b (DE between G1 + G2 and G3) and miR-let-7b (DE between G1 and G2 + G3) were discovered in our cohort as well (Appendix A). With this signature, the authors showed that patients in a high-risk group have poorer OS compared to the low-risk group. Unfortunately, this classification rule was less efficient than the clinical stage but was superior to histological type or neoplasm histological grade [42]. The potential of miRNA profile in predicting the course of the disease is high, however, the use of miRNA-based stratification should be assisted with genomic subtyping. We expect that miRNA profiling may become a useful tool in the future to predict a patient’s prognosis and survival even independently from the current grading system based on histopathological results.

Specific processes that reflect the current state of cancer cells (tumor tissue) at the time of collection are associated with the miRNA profile. Based on this profile, we attempted to find a process responsible for inhibiting cell differentiation. We confirmed that specific miRNAs target genes that are responsible for negative regulation of cell differentiation (GO:0045596) toward a higher grade. However, this result is rather a verification (or validation) of the correctness of the study, as well as the “capacity” of miRNA in the sense of marker of tumor biology. It was surprising to find a process of modulation by viruses of host morphology or physiology. Current knowledge about the pathogenesis of EC excludes the possible role of HPV virus in its etiology, despite detecting its presence (HPV 16–24% and HPV 18–20%) in postsurgical material [43,44]. However, other viruses such as human mammary tumor virus (HMTV) or mouse mammary tumor virus-like virus were also present EC tumors [45,46]. Therefore, it would be valid to investigate other tumorigenic viruses in high grade EC.

## 5. Conclusions

1. NGS profiling of miRNAs enhances our understanding of endometrial cancer biology and allows the discovery of new biomarkers;

2. Expression of selected miRNAs can distinguish each EC grade with less error likelihood compared to morphological examination;

3. The level of miRNA expression heterogeneity of grade 2 EC samples is much higher than expected.

## Figures and Tables

**Figure 1 biomolecules-11-00274-f001:**
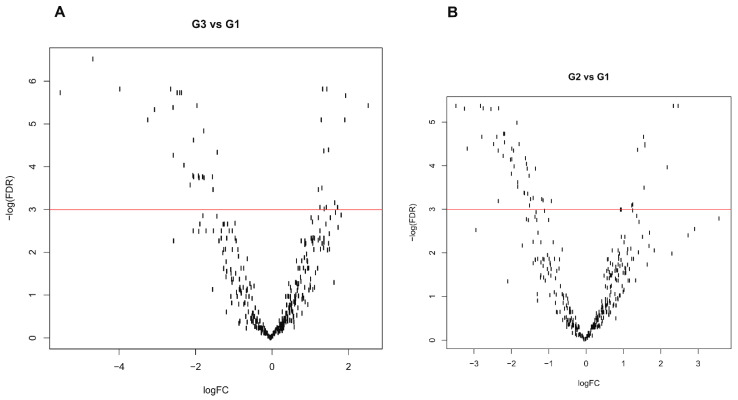
Volcano plots of differential expressed genes (DEG) between individual grades. (**A**) DEG between grade 3 and grade 1 samples. (**B**) DEG between grade 2 and grade 1 samples. *X* axis, logarithmic fold change value (logFC) and *Y* axis, negative logarithmic value of false discovery rate (−logFDR). Each dot represents an individual miRNA transcript.

**Figure 2 biomolecules-11-00274-f002:**
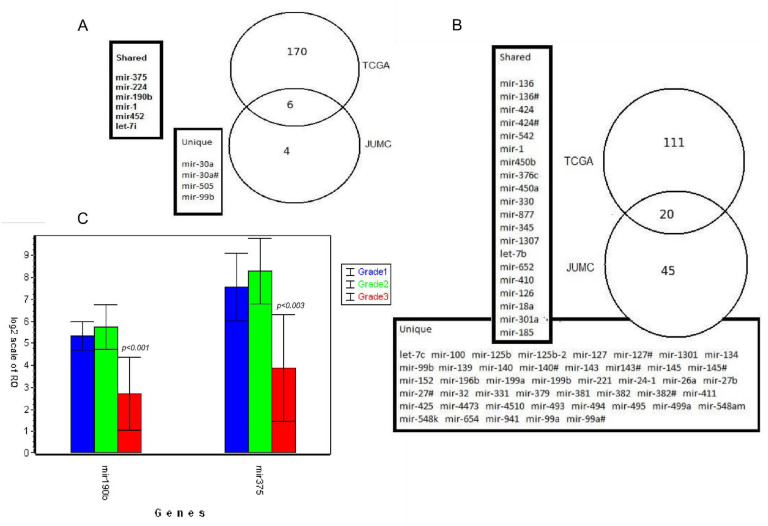
Overlapping miRNA genes between studied cohort (JUMC) and data collected by the Cancer Genome Atlas (TCGA) cohort. Venn diagrams shows the number of differentially expressed genes between G1 + G2 vs. G3 (**A**) and G1 vs. G2 + G3 (**B**). Unique and shared miRNA are listed in frames. (**C**) shows qPCR relative expression results of miR-190b and miR-375. JUMC, Jagiellonian University Medical College); studied cohort, TCGA; the Cancer Genome Atlas; blue bar, grade 1; green bar, grade 2; red bar, grade 3.

**Figure 3 biomolecules-11-00274-f003:**
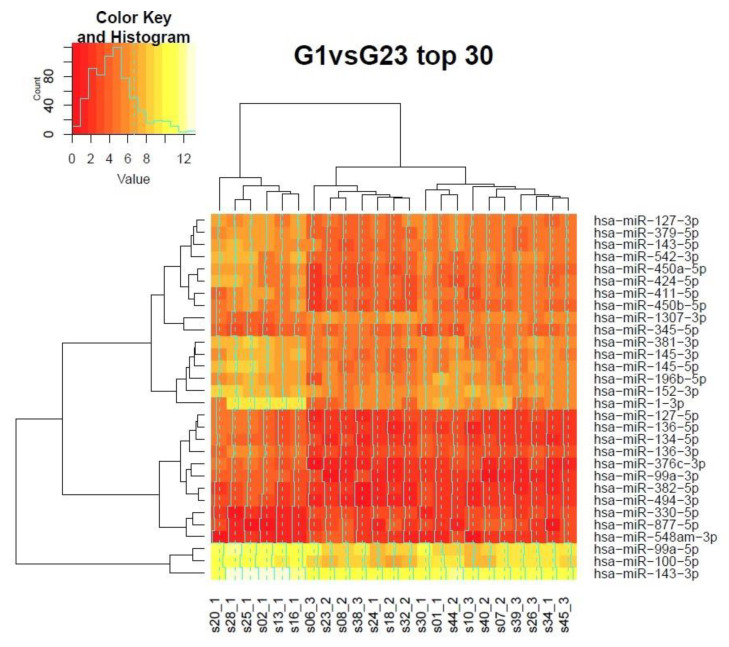
Clustering of the top 30 differentially expressed miRNAs of G1 patients vs. collected-together patients with G2 and G3 grade. *Value* = miRNA gene reads. At the bottom of the picture is the sample’s name with grade indication.

**Figure 4 biomolecules-11-00274-f004:**
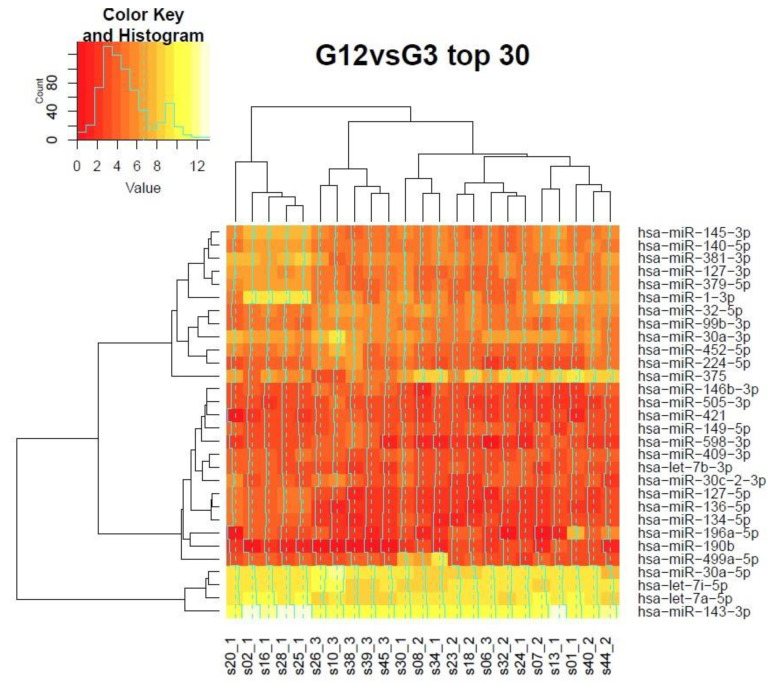
Clustering of the top 30 differentially expressed miRNAs of collected together patients with G1 and G2 grade vs. G3 grade (low grade vs. high grade EC). Clustering includes miR-190b and miR-375. It can be seen that miR-375 clusters differently from other miRNAs. *Value* = miRNA gene reads. At the bottom of the picture is the sample’s name with grade indication.

**Table 1 biomolecules-11-00274-t001:** Phenotypic data for each grade. Median with range (in brackets) was calculated for each complex trait.

Phenotype	Grade 1	Grade 2	Grade 3	Wilcoxon Test (*p*-Value)
Age (years)	66 (54–78)	54 (53–65)	64 (46–78)	*0.1099*
Height (cm)	163 (150–170)	164 (158–170)	165 (156–168)	-
Weight (kg)	80 (63–100)	74 (53–108)	67 (52–98)	-
No. of menstruations	375 (340–430)	380 (350–410)	350 (290–430)	*0.6508*
Time since last menstruation (months)	51.5 (48–57)	53 (49–53)	50 (44–57)	*0.5787*
Pregnancies	2 (0–4)	2 (1–4)	2 (1–3)	*0.6466*
BMI	29.04 (23.31–40.90)	28.04 (20.70–40.15)	24.61 (20.57–38.21)	*0.4982*

**Table 2 biomolecules-11-00274-t002:** Confusion matrix of the 10 most discriminative miRNAs. Classification error indicates the maximum likelihood of making an error in the case of a particular grade.

Grade vs. Grade	1	2	3	Class. Error	miRNAs
1	9	1	0	0.1	hsa-miR-652-3p, hsa-miR-4510,hsa-miR-190b, hsa-miR-421,hsa-miR-941,hsa-miR-1307-3p, hsa-miR-375,hsa-miR-345-5p, hsa-miR-99b-3p, hsa-let-7b-3p
2	1	6	0	0.1429	
3	0	1	6	0.1429	

## Data Availability

Generated data are publicly available in the text and in supplementary information files. Raw data as well as bioinformatic code are available from corresponding author on request.

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
