# Peer review of "Low Expression of miR-375 and miR-190b Differentiates Grade 3 Patients with Endometrial Cancer"

_biomolecules, 2021, doi:10.3390/biom11020274_

Round 1

Reviewer 1 Report

A manuscript, entitled “Low expression of miR-375 and miR-190b differentiates grade 3 patients with endometrial cancer“ by Pietrus, M. et al. is devoted to a search for new diagnostic and prognostic markers for endometrial cancer (EC).

An idea of a work is very important and interesting, because expression levels of two miRNA - miR-375 and miR-190b – differs in grade 3 tumor samples, than in G1 and G2. I was impressed by nice combination of experimental approach, a bioinformatics analysis and statistics.

This is very important for the individualized treatment of EC patients.

I have few comments, anyway, listed below:

  1. Figure 1 is not referred in the body text; Figure 2 is confused with Figure 1. Please, check this carefully.
  2. There is no Figure 5.
  3. On Figure 2 there are no labels, as they should be.
  4. It is not clear, what samples were pooled per run and why? If the authors had only 7, 7 and 10 of samples of each grade, how these samples were merged? Or the authors did not explain well this point. Please, clarify this.
  5. Why I can see a line about viruses in an abstract, if only one sentence in the end of this manuscript is devoted to HPV, referring to 2 review papers?
  6. In continuation of p.5, what other viruses, than HVP were ever shown to be associated with EC?
  7. I do not think that only expression of these two miRNAs (miR-375 and miR-190b) should be used for diagnostics. For sure, other markers should be used in the parallel.

Concluding, I would suggest to accept the present manuscript after revision.

Reviewer 2 Report

Endometrial cancer is the most common gynecological cancer in the Western world and the fourth most common cancer in women globally [1], with over 380,000 new cases and 89,000 deaths in 2018 [2] This pathology has shown an increase in both incidence in the last decade that, even more, of mortality due to imprecise guided patient management from histopathology [1-5].

The current histopathological risk assessment is indeed poorly reproducible, leading to excessive or insufficient treatment of women e misinterpretations of results in clinical trials [6,7]. As prognosis does not directly correspond to the grading and histotype, the objective of the study is very relevant.

Specific comments:

Abstract:

In our opinion, the abstract is clearly structured, explaining the purpose of the study and the

conclusions very well.

However, not only grade 3 requires adjuvant therapy, as grades 1 and 2 may also require adjuvant brachytherapy and radiotherapy, according on the stage of the disease.

Introduction:

In line 38 is not right to say that grade 1 and 2 are treated in the same way, as grade 1 surgical treatment may not include pelvic and obturator lymphadenectomy and in many cases grade 1 may not require adjuvant therapy.

In line 47 it would be correct to specify that surgical treatment with hysterectomy alone is performed in the early stages, considering also the myometrial infiltration margins.

Material and methods:

it is not clear whether any familiarity was investigated among the patients and if this was an exclusion criteria.

Was the population homogeneous? please specify if it was comparable among different groups.

Discussion:

The discussion makes it clear what the potentials and limits of the study are.

Conclusion:

Further prospective studies are needed to provide more accurate answers, although this paper may help better stratify endometrial cancer to improve survival.

[1] Siegel RL, Miller KD, Jemal A. Cancer statistics, 2015. CA Cancer J Clin. 2015 Jan-Feb;65(1):5- 29.

[2] Bray F, Ferlay J, Soerjomataram I, Siegel RL, Torre LA, Jemal A. Global cancer statistics 2018: GLOBOCAN estimates of incidence and mortality worldwide for 36 cancers in 185 countries. CA Cancer Journal for Clinicians. 2018;68(6):394-424.

[1] Siegel RL, Miller KD, Jemal A. Cancer statistics, 2015. CA Cancer J Clin. 2015 Jan-

Feb;65(1):5- 29.

[2] Bray F, Ferlay J, Soerjomataram I, Siegel RL, Torre LA, Jemal A. Global cancer

statistics 2018: GLOBOCAN estimates of incidence and mortality worldwide for 36 cancers

in 185 countries. CA Cancer Journal for Clinicians. 2018;68(6):394-424.

[3] Talhouk A, McConechy MK, Leung S, et al. A clinically applicable molecular-based classification for endometrial cancers. Br J Cancer. 2015;113(2):299-310.

[4] Giampaolino P, Di Spiezio Sardo A, Mollo A, et al. Hysteroscopic Endometrial Focal Resection followed by Levonorgestrel Intrauterine Device Insertion as a Fertility-Sparing Treatment of Atypical Endometrial Hyperplasia and Early Endometrial Cancer: A Retrospective Study. J Minim Invasive Gynecol. 2018 Jul 11. pii: S1553-4650(18)30347-9.

[5] Gilks CB, Oliva E, Soslow RA. Poor interobserver reproducibility in the diagnosis of high-grade endometrial carcinoma. Am J Surg Pathol. 2013;37:874-881.

[6] Hoang LN, McConechy MK, Kobel M, et al. Histotype-genotype correlation in 36 high grade endometrial carcinomas. Am J Surg Pathol. 2013;37:1421-1432.

[7] Talhouk A, McConechy MK, Leung S, et al. Confirmation of ProMisE: A simple, genomics-based clinical classifier for endometrial cancer. Cancer. 2017;123(5):802-813.

Reviewer 3 Report

The study by Pietrus et al. raises an important clinical problem, which is the differentiation between low-grade endometrial cancer (Grade 1-2), with good prognosis, and highest grade (Grade 3), with poor prognosis. The authors propose to use the assessment of small non-coding RNA molecules (i.e. miRNA) and indicated that the concentration of both miR-375 and miR-190b differentiates EC grades 1-2 from EC grade 3. Despite the small sample size, the study still has significant clinical value. I have only a few minor comments:

  1. The Abstract does not refer to the main result, i.e. miR-375 and miR-190b.
  2. To emphasize the importance of the problem, in the Introduction it is worth paying attention on the epidemiological data. Despite the fact that EC is usually diagnosed at the early stages and gives satisfactory treatment results, it is worth referring at least to aspects, such as, the average 5-year survival time, differentiating between Grade 1-2 and Grade 3.
  3. In several places there is no reference to the literature, e.g. a line 57, 301…
  4. Lines 68-75, the authors emphasize that, “unfortunately”, most research on miRNA expression has so far been carried out via qPCR or microarrays, but not NGS. Please briefly explain why these techniques are of less clinical value than NGS as authors implicated. Elaboration on this observation will have an interesting value for readers.
  5. The age range in Table 1 does not correspond to the age range indicated in the Materials and Methods section.
  6. Figure 1 and Figure 2 are unreadable and requires improvement.
  7. Figure 2c lacks description of the Y axis and there is no indication of statistical data. In Figure 2, I did not find any references to the abbreviations posted under the figure, CMUJ or TCGA.
  8. The results reported in lines 230-232 regarding miR-375 and miR-190b should have statistical support as they constitute a significant element to which the authors refer in the Title of the paper. It is not enough to mention and describe "data not shown".
  9. Currently, in the results of the study, the aspect emphasized in the Title of the paper is slightly underlined.
  10. In the Discussion section, the authors should emphasize the novelty and originality of their study. It is worth emphasizing what this study found “for the first time”.
  11. Line 285: please provide for summary how miR-375, miR-214, and miR190b differentiates Grades 1-2 from Grade 3.
  12. The Discussion section needs improvement and more attention should be given to discussing the results obtained. Currently it is too general. Moreover, what do we know about miR-375 and miR-190b?
